# Semiconductor-to-Insulator Transition in Inter-Electrode Bridge-like Ensembles of Anatase Nanoparticles under a Long-Term Action of the Direct Current

**DOI:** 10.3390/nano13091490

**Published:** 2023-04-27

**Authors:** Dmitry A. Zimnyakov, Sergey S. Volchkov, Mikhail Yu. Vasilkov, Ilya A. Plugin, Alexey S. Varezhnikov, Nikolay V. Gorshkov, Arseni V. Ushakov, Alexey S. Tokarev, Dmitry V. Tsypin, Dmitry A. Vereshagin

**Affiliations:** 1Physics Department, Yury Gagarin State Technical University of Saratov, Saratov 410054, Russia; volchkov93@bk.ru (S.S.V.); vasilk.mikhail@yandex.ru (M.Y.V.); ilyaplygin@mail.ru (I.A.P.); alexspb88@mail.ru (A.S.V.); gorshkov.sstu@gmail.com (N.V.G.); arhangel92r@mail.ru (A.S.T.); qamadeusp@gmail.com (D.V.T.); verechagin2011@mail.ru (D.A.V.); 2Precision Mechanics and Control Institute of Russian Academy of Sciences, 24 Rabochaya St., Saratov 410024, Russia; 3Saratov Branch of Kotelnikov Institute of Radioengineering and Electronics of RAS, Saratov 410019, Russia; 4Chemistry Department, Saratov State University, 83 Astrakhanskaya St., Saratov 410012, Russia; arsenivushakov@ya.ru

**Keywords:** conductivity, anatase nanoparticles, inter-electrode bridges, percolation threshold, critical exponent

## Abstract

The results of experimental studies of ohmic conductivity degradation in the ensembles of nanostructured anatase bridges under a long-term effect of direct current are presented. Stochastic sets of partially conducting inter-electrode bridges consisting of close-packed anatase nanoparticles were formed by means of the seeding particles from drying aqueous suspensions on the surfaces of silica substrates with interdigital platinum electrodes. Multiple-run experiments conducted at room temperature have shown that ohmic conductivity degradation in these systems is irreversible. It is presumably due to the accumulated capture of conduction electrons by deep traps in anatase nanoparticles. The scaling analysis of voltage drops across the samples at the final stage of degradation gives a critical exponent for ohmic conductivity as ≈1.597. This value satisfactorily agrees with the reported model data for percolation systems. At an early stage of degradation, the spectral density of conduction current fluctuations observed within the frequency range of 0.01–1 Hz decreases approximately as 1/ω, while near the percolation threshold, the decreasing trend changes to $\approx 1/\omega^{2}$. This transition is interpreted in terms of the increasing contribution of blockages and subsequent avalanche-like breakdowns of part of the local conduction channels in the bridges into electron transport near the percolation threshold.

## 1. Introduction

In recent years, stochastic and quasi-regular structures based on the close-packed semiconductor nanoparticles, assemblies of nanoparticles with carbon nanofibers, and dielectric, conductive, and semiconducting nanosheet networks have been the object of particular attention as material platforms to be applied in sensorics [1,2,3,4], photonics [5,6,7,8,9], catalytic chemistry [10,11,12], energy storage [13,14], printed electronics [15,16], etc. In addition to various applications, investigation of the characteristics of charge and photon transfer at microscopic, mesoscopic, and macroscopic levels of these systems can contribute to the further development of such fundamental areas of modern science as the percolation theory, physics of complex non-stationary systems, solid state physics, optics of random media, etc. Among the variety of nanostructured semiconductor materials that can be used to create systems controlled by the light or external electric field, indirect-gap semiconductors characterized by strong interaction of conduction electrons with a crystal lattice are of particular interest. Combined with a high concentration of surface traps in the nanoparticles, these features will lead to long-term responses of the systems to external electrical or optical influence with low relaxation rates, which can be interpreted as memory effects in such structures. In turn, this opens a way to create bistable functional nanomaterials to be applied in sensorics, photonics, electronics, and photo-catalytic chemistry.

Anatase is a typical representative of such materials, being a wide-gap n-type semiconductor with an indirect interband transition. The reported bandgap values for the bulk anatase are around $E_{g}\approx$ 3.2 eV [17,18]. A remarkable influence of the electron–phonon coupling on the charge transport is characteristic of this material [19,20]. This leads to a high probability of the occurrence of polaron-type conductivity and, accordingly, to a relatively low mobility of conduction electrons. In addition, the presence of a large number of shallow traps increases the role of hopping conductivity in the transport of mobile charge carriers in anatase nanoparticles. Z-scan experiments with water suspensions of anatase nanoparticles irradiated by a pulse-periodic high-intensity laser light in the fundamental absorption band (at 355 nm, [21,22]) show the featured behavior of an effective dielectric function of nanoparticles under long-term irradiation. Such behavior can be interpreted as a transition from a “semiconducting” state of an anatase nanoparticle to an “insulating” state and manifests itself in a dramatic decay of the modulation depth of the time-averaged effective dielectric function with an increase in the duration of pulse-periodic laser action. This effect can be explained in terms of a joint influence of the low recombination rate of photo-induced charge carriers, the high probability of trapping mobile carriers, and the depletion of the ground state of anatase molecules in the nanoparticles [22].

Localization of mobile charge carriers arrested by the traps combined with depletion of ensembles of the major charge carriers in the particles can also play a crucial role in long-term macroscopic DC charge transport through the systems of densely packed anatase nanoparticles. It should be expected that a gradual decrease in the volume-averaged concentration of mobile carriers in these systems will lead to significant changes in their macroscopic conductivity over time. An extreme state of the system, achievable as a result of such evolution, is an “insulator” state, due to the close-to-zero average concentration of free charge carriers in the volume of the system.

The goal of this work was an experimental study and interpretation of this phenomenon in stochastic low-dimensional anatase nanostructures, which are random sets of local inter-electrode bridges and are subjected to long-term exposure to direct electric current.

## 2. Materials and Methods

### 2.1. Sample Preparation and Characterization

The studied stochastic structures of aggregated anatase particles were formed by “seeding” nanoparticles onto the surfaces of specially prepared silica substrates with flat interdigital electrode systems within the working zones overlapped by electrodes. These platinum interdigital electrode systems were produced on the surfaces of silica plates with the size of 9.0 × 10.0 mm^2^ and thickness of 0.3 mm using PVD technology (see Figure 1a as an example of the substrate design). To expand the functionality of the substrates, they also contained flat resistive heaters and flat thermoresistors (respectively, positions H and TR in Figure 1a). Two types of interdigital electrode systems with the following geometrical parameters were used:(1)with an inter-electrode distance of (103.8 ± 3.9) μm and electrode width of (89.8 ± 5.7) μm;(2)with the inter-electrode distance of (44.9 ± 4.0) μm and electrode width of (54.2 ± 5.2) μm.

In both cases, the number of electrode pairs was equal to 19; the sizes of the working zones defined by electrode-covered areas were 4.29 × 7.37 mm^2^ in the first case and 4.26 × 3.84 mm^2^ in the second case. The heights of electrodes above the surfaces of the substrates, measured using the Dektak 150 profilometer (the product of Veeco, Plainview, NY, USA), were approximately (0.88 ± 0.02) μm in the first case and (1.61 ± 0.01) μm in the second case. Before electrical measurements, the contact pads of even and odd electrodes on the sides of the substrates were interconnected by the jumpers and led out to connect them with the measuring units. 

“Seeding” of TiO_2_ anatase nanoparticles onto the surfaces of substrates was carried out by applying small volumes of aqueous suspensions of particles to preliminarily cleaned substrate surfaces, followed by slow evaporation of water from the surfaces. The anatase nanopowder produced by Sigma Aldrich Inc., Burlington, MA, USA (item #637254 in the product list, powder of polydisperse anatase nanoparticles with an average size ≤ 25 nm, [23]) was applied as the basic component of the prepared suspensions. Before preparing suspensions, the anatase powder was inspected using the X-ray diffractometry (DRON-8T X-ray diffractometer, the product of Bourevestnik, JSC, Saint-Petersburg, Russia) and the PDF-2 database (ICDD, Newtown Square, PA, USA). To record diffraction patterns, we used CuKα radiation, the Goebel parabolic mirror (AXO Dresden GmbH, Dresden, Germany), and the Mythen 2R1D position-sensitive detector (DECTRIS AG, Baden-Daettwil, Switzerland). Figure 2 displays the X-ray diffractogram for the examined powder sample, and the design curve according to the full profile analysis. Based on the results of the full profile analysis (FullProf Suite), the anatase phase is characterized by the parameters of the tetragonal unit cell: a= (3.78608 ± 0.00009) Å and c= (9.50642 ± 0.00025) Å. For comparison, the database values (PDF no. 00-021-1272) are equal to 3.7852 Å and 9.5139 Å, respectively. Note that the average size of crystallites in the powder estimated using the Williamson–Hall technique [24,25] (≈(23 ± 2) nm) fairly agrees with the upper level of the average size specified by the producer.

Suspensions were prepared using deionized water with the mass fraction of particles varied from 0.2 μg/mL to 0.5 μg/mL. The volume of suspensions applied to the surface of the substrates was equal to 8 μL. Note that in the case of an ideal uniform distribution of the sown particles across the working areas of 4.26 × 3.84 mm^2^, the mass fraction of 0.2 μg/mL corresponds to a particle layer with a thickness of ≈25 nm. In practice, the formation of examined structures is governed by a variety of stochastic processes at the mesoscopic level, including local inhomogeneities in evaporation rates and capillary forces. Accordingly, long-term evaporation of water (approximately 2 h duration) from the suspension layers at room temperature led to the formation of random distributions of single and aggregated nanoparticles across the working areas with a dominating occurrence of aggregated particles along the initial borders of the applied drops of suspensions (the coffee ring effect, Figure 1b). Part of aggregated particles formed conductive bridges between adjacent electrodes, which can be identified by microscopic inspection of the surface of the fabricated samples (Figure 1c). The number of these bridges providing initial conductivity of the studied samples ranging from 10^−9^ S to 10^−8^ S was approximately from 50 to 300 with an average width from 15 to 45 μm (depending on the mass fraction of particles in the applied drops). The heights of the bridges above the substrate surfaces were several dozen nanometers; five arbitrarily chosen bridges were traced using the Dektak 150 profilometer at the minimal value of the stylus force (3 mg). Figure 3 displays a typical smoothed profilogram for one of the examined items. Based on the analysis of obtained smoothed profilograms, it was found that the following approximating relationship between the width w and average height $\bar{h}$ of bridges (see Figure 3) can be used with acceptable accuracy: $\bar{h}\approx K_{b}w$, where $K_{b}\approx$ (4.80 ± 0.32) × 10^−3^. 

Due to stochasticity and, accordingly, uncontrollable formation of sets of inter-electrode bridges in the “seeding” process, all the produced samples were subjected to thorough microscopic inspection before measuring their conductive properties. The inspection was carried out using the Olympus MX51 optical microscope in reflected light in the dark field mode (due to the high reflectivity of the substrate surfaces). With the used Olympus MPLFLN150xBD objective, the magnification was 1500×. Images of the surface fragments were captured over all the areas of the inspected samples with the resolution of 2448 × 1920. Panoramic images of the samples presented in Figure 1a,b were obtained with significantly lower magnification (50×) when using the Olympus UIS2 MPLFLN5xBD objective. 

In the course of the inspection, the processing of the fragments of microscopic images with the inter-electrode bridges was carried out in the following way (Figure 4): the processed fragment (a) was converted to a black-and-white format (b). The conversion was carried out in the interactive mode using a manual tracing of the bridge boundaries and follow-up automatic filling of the area between the traced boundaries by black pixels. When tracing the bridge boundaries, adjacent island structures, and narrow and long branches oriented mainly across the inter-electrode electric field were excluded (see Figure 4b). These fragments should not have a significant effect on the transfer of charge carriers through the bridges. On the other hand, the existing voids inside the bridges were also contoured in order to adequately assess the area covered by the bridges.

After this, the bridge width was estimated as $w\approx N_{bp}S_{p}\sin\phi /d_{ie}$, where $N_{bp}$ is the number of automatically counted black pixels, $S_{p}$ is the area covered by a single pixel in the image, $d_{ie}$ is the inter-electrode distance, and ϕ is the inclination angle of the bridge with respect to electrodes (Figure 4b). Calculation of $N_{bp}$ in each binarized image of the bridges was carried out using specially developed software in the Matlab environment. 

As a result of inspection, the sets of width values $\left\{w_{i}\right\}$ were obtained for all the produced samples. Statistical analysis of these datasets showed that sample probability density functions of w values with acceptable accuracy allow approximation by the lognormal distribution:(1)$$\rho \left(w\right)=\rho_{0}+\frac{1}{\sqrt{2\pi }w\sigma_{w}}\exp\left(-\frac{{\left(\ln\left\{w/w_{c}\right\}\right)}^{2}}{2\sigma_{w}^{2}}\right),$$
where $\sigma_{w}$ and $w_{c}$ are distribution parameters ($\langle w\rangle =w_{c}\exp(\sigma_{w}^{2}/2)$), and $\rho_{0}$ is the offset occurring in the approximation. Figure 5 presents examples of histograms of w values along with approximating lognormal distributions for two samples with significantly differing average widths $\langle w\rangle$ of inter-electrode bridges. In a further study, the values of $\langle w\rangle$ and the number $N_{b}$ of bridges connecting adjacent electrodes were specified for each examined sample.

### 2.2. Sample Examination

During the main experiments, time-dependent values of the ohmic conductivity G in the fabricated samples were studied in the direct current mode with the constant current of 1.0 × 10^−9^ A. Accordingly, instantaneous values of the voltage drop across the samples were acquired along increasing time lapses; Figure 6 displays the corresponding experimental arrangement.

The voltage-controlled current source SRS CS-580 from Stanford Research Systems (item 1) was applied to provide the established direct current in the examined samples. Instantaneous values of the voltage drop $U\left(t\right)$ across the sample (item 3) were recorded using the data acquisition system based on the multimeter Agilent 34401A (item 2) controlled by PC. The sampling frequency was set equal to 20 Hz, and each recording run continued until the voltage drop across the examined sample reached 20 V. When this voltage threshold was overcome, the current source switched to the voltage stabilization mode, and the data recording run was stopped. This limiting value was chosen to exclude possible electrical breakdowns of interelectrode gaps with a further increase in the voltage drop. Some of the samples were subjected to multiple (up to 6 times) sequential runs with 90 min intervals between them. The purpose of such repeated experiments with the same samples was to evaluate the recovery rate for their conductivity, which dramatically fell after the completion of the first runs. The experiments were carried out at room temperature (25 °C).

During the experiments, the capacitance of the substrates with electrode systems and produced samples were measured before the formation of ensembles of inter-electrode bridges ($C_{substrate}$), before the start of the first runs ($C_{sample}$), and after their completion (${C^{\prime }}_{sample}$). In addition, the capacitance of the connecting cables (the parasitic capacitance, $C_{par}$) was measured; measurements were provided using a multimeter unit (Keithley DAQ6510 type) with an error no worse than ±1.0 pF. Additional measurements of these capacitance values using the Novocontrol Alpha AN impedance measuring system (Novocontrol Technologies GmbH & Co. KG, Montabaur, Germany) in the frequency range from 0.01 Hz to 10^5^ Hz are in good agreement with the results obtained with the Keithley DAQ6510 multimeter.

Typically, the $C_{substrate}$ value for the first type of substrates is ≈(490 ± 5) pF; the substrates of the second type (with inter-electrode gaps of ≈44.9 μm) are characterized by the capacitance of (215 ± 3) pF. The formation of ensembles of inter-electrode bridges causes a small but systematic increase in the capacitance of the examined samples $C_{sample}-C_{substrate}\approx$ (5 ÷ 7) pF, depending on the number and average width of the formed bridges). After the first runs, the differences ${C^{\prime }}_{sample}-C_{substrate}$ decrease with respect to $C_{sample}-C_{substrate}$ down to the values of the order (2 ÷ 3) pF. The reason for this behavior is considered in Section 4. Accordingly, these insignificant changes in the capacitance of the samples allow us to assume that $C_{sample}\approx const$ in the course of the first and subsequent runs. 

The measured parasitic capacitance was approximately equal to ≈180 pF; in combination with $C_{sample}$, this value was used to obtain the total capacitance $C_{tot}=C_{sample}+C_{par}$ necessary for the recovery of the current values of the total ohmic conductivity G for the ensembles of inter-electrode bridges. 

## 3. Results

Figure 7a illustrates a typical behavior of the voltage drops across the examined samples with various values of $\langle w\rangle$ and $N_{b}$ during the first runs. Dependences $U\left(t\right)$ can be divided into three characteristic regions: a short-term transitional region with a duration of the order of several seconds (I); a quasi-stationary region, characterized by a slow increase in the voltage drop (II); and a region of rapid increase in $U\left(t\right)$(III). The duration of the quasi-stationary region II abruptly increases with an increase in the number of inter-electrode bridges $N_{b}$ in the sample. In particular, pilot experiments with the samples fabricated with high mass fractions of nanoparticles in the applied suspensions (of the order of 0.7 μg/mL or more) and, correspondingly, characterized by large values of $N_{b}$ (of the order of 300 or more), showed absence of the “II → III” transition at the run times of the order of 6 h or more (as an example, see the inset in Figure 7b). It can be assumed that this feature is due to the existence of a certain critical value for the average current $I/N_{b}$ through the inter-electrode bridges. When this critical value is exceeded in the ensemble of the bridges, a rather rapid degradation of their conducting structure occurs and, as a result, the “II → III” transition takes place within a foreseeable time.

Figure 7b displays a typical set of $U\left(t\right)$ dependences obtained in the experiment with the same sample subjected to a series of sequential runs with 90 min intervals between them. It should be taken into account that in the case of direct current through the sample, a linear increase in the voltage drop $U\left(t\right)\propto t$ corresponds to its dominant capacitive susceptance. On the contrary, the case of $U\left(t\right)\approx const$ indicates a prevailing ohmic conductance in the sample. Accordingly, the data set shown in Figure 7b indicates a catastrophic depletion of ensembles of free charge carriers in the inter-electrode bridges in sequences of runs with low efficiency in restoring their concentration between runs.

The dependencies $U\left(t\right)$ obtained along the first runs were pre-processed to separate them into the trend ($\bar{U}\left(t\right)$) and fluctuation ($U_{f}\left(t\right)$) components using the adjacent averaging procedure; a running window with a width of 100 s was applied. As an example, Figure 8, displays the result of this preprocessing for sample 3 (see Figure 7). Consideration of the sampled power spectra of fluctuation components calculated for 100 s sampling intervals within regions II and III showed that they allow the power-law approximation $S_{U_{f}}\left(\omega \right)\propto \omega^{-\gamma^{\prime }}$ with acceptable accuracy in the frequency intervals of 0.01 Hz ÷ 1 Hz (Figure 8b). A remarkable feature is that, despite significant differences in the sampled power values $\bar{U_{f}^{2}\left(t\right)}$ of fluctuation components for various samples at stages II and III, the values of the spectral exponents $\gamma^{\prime }$ for all samples exhibit a general tendency to increase from the values of the order of (2.6 ÷ 3.1) to the values of (3.5 ÷ 4.0) in the “II–III” transition. The reason for this behavior is considered in the Section 4.

## 4. Discussion

### 4.1. Recovery of Time-Dependent Smoothed Conductivity in the Examined Samples

The basic relationship describing the voltage response $U\left(t\right)$ of the examined samples under the action of the constant direct current I can be written in the following form:(2)$$I=G\left(t\right)U\left(t\right)+C_{tot}\frac{dU\left(t\right)}{dt}.$$

Here we neglect small variations of the total capacitance of the system “sample + connecting cables” along the runs. Considering the trend components $\bar{U}\left(t\right)$ of the “voltage drop-time lapse” dependencies, we arrive at the following expression for evaluation of the smoothed time-dependent conductivity:(3)$$\bar{G}\left(t\right)=\frac{I-C_{tot}\left(d\bar{U}\left(t\right)/dt\right)}{\bar{U}\left(t\right)}.$$

In this case, the term “smoothed conductivity” means evaluation of G at the moment using the corresponding values of the voltage and its first derivative estimated for the trend component $\bar{U}\left(t\right)$. As an example, Figure 9 displays the values of $\bar{G}$ against the time-lapse recovered for the series of sequential runs (for the initial datasets see Figure 7b). The recovered dataset $\bar{G}\left(t\right)$ for the first run exhibits the behavior at the final stage of the run, which is typical for percolation systems near the threshold.

The horizontal dashed line corresponds to the minimal value of “pure” ohmic conductivity (5 × 10^−11^ S) achieved under $\bar{U}\left(t\right)$ = 20 V in the case of $C_{tot}\left(d\bar{U}\left(t\right)/dt\right)$ = 0. During the first runs, the achievable maximal values of the displacement current in the samples are many times less than the set value of the total current (1 nA). On subsequent runs, the contribution of the displacement current to the total current gradually becomes dominant. Accordingly, the recovered $\bar{G}$ values are below the dotted threshold line on the graph.

The arrows indicate partial recovery of ohmic conductivity in the examined sample during 90 min time intervals between the first-second and second-third runs. However, it should be noted that this recovery under the used conditions (in particular, at room temperature) is subtle and decreases from run to run. In the subsequent runs, ohmic conductivity occurs vanishingly small (the single cyan marker). 

### 4.2. Scaling Behavior of $\bar{G}$ and the Conductivity Critical Exponent

At the final stages of the first runs, when conducting systems in the studied samples approach their critical states, contributions of the displacement current $C_{tot}\left(d\bar{U}\left(t\right)/dt\right)$ to the total current are small and the trend components $U\left(t\right)$ are mainly governed by $\bar{G}\left(t\right)$: $\bar{U}\left(t\right)\approx I/\bar{G}\left(t\right)$. On the other hand, the general property of percolation systems is the power-law dependence of the system conductivity on detuning the governing parameter $p\left(t\right)$ from its critical value $p_{c}$: $\bar{G}\left(t\right)\propto {\left[\Delta p\left(t\right)\right]}^{\alpha }$, where $\Delta p\left(t\right)=\left[p_{c}-p\left(t\right)\right]/p_{c}$. For the examined systems, which consist of large sets of statistically independent random conduction channels (percolation clusters), the control parameter $p\left(t\right)$ is associated, for example, with the number of blocked nodes or bonds in these percolation clusters. Accordingly, when $\Delta p\left(t\right)\to 0$, $\bar{G}\left(t\right)$ falls (α should have a positive value). The trend component of the voltage drop $\bar{U}\left(t\right)$ should exhibit a similar behavior near the threshold of ohmic conductance: $\bar{U}\left(t\right)\propto {\left[\Delta p\left(t\right)\right]}^{\beta }$ (β=−α is negative). The first derivative $d\bar{U}\left(t\right)/dt$ can be expressed as $d\bar{U}\left(t\right)/dt\propto {\left[\Delta p\left(t\right)\right]}^{\beta -1}\left[d\left\{\Delta p\left(t\right)\right\}/dt\right]$. This expression can be rewritten in the following form: $d\bar{U}\left(t\right)/dt\propto {\left[\bar{U}\left(t\right)\right]}^{\frac{\beta -1}{\beta }}\left[d\left\{\Delta p\left(t\right)\right\}/dt\right]$. We can assume that, at a rather short stage of rapid growth in $\bar{U}\left(t\right)$, the dependence $p\left(t\right)$ admits a linear approximation ($p\left(t\right)\propto t$ and, accordingly, $d\left\{\Delta p\left(t\right)\right\}/dt\approx const=K_{p}$. Under these assumptions, we arrive at the following reduced relationship between the current values of $\bar{U}\left(t\right)$ and corresponding first derivatives:(4)$$\frac{\bar{U}\left(t\right)}{dt}\propto {\left[\bar{U}\left(t\right)\right]}^{\frac{\beta -1}{\beta }}K_{p}.$$

Figure 10a illustrates the procedure for estimating $\bar{U}$ and $d\bar{U}/dt$ values from the obtained trend curves $\bar{U}\left(t\right)$. Figure 10b displays in the logarithmic coordinate the sets of pairs $\left(d\bar{U}/dt,\bar{U}\right)$ corresponding to the final stages of the first runs for various examined samples. Note that, despite a remarkable scattering of these datasets across the $\left(d\bar{U}/dt,\bar{U}\right)$ coordinate plane, they exhibit common power-law trends with the close values of slopes in logarithmic coordinates. The red dashed line with a slope of ≈0.615 ± 0.029 indicates this common trend for all the datasets and serves as a guide for the eye. Selectively shown error bars for the values of $d\bar{U}/dt$ correspond to the confidence level of 0.9. Note that for this data presentation scheme ($d\bar{U}/dt$ is an argument, and $\bar{U}$ is a function), the slope of the trend line is determined by the ratio $\beta /\left(\beta -1\right)$; thus, the critical exponent for the trend value of the voltage drop is estimated as ≈−1.597.

According to Equation (4), mutual shifts of the datasets in the coordinate plane can be due to different values of the parameter $K_{p}$ for these samples. As follows from the above consideration, this parameter characterizes the rate in the increase of the number of blocked nodes or bonds in the model percolation system that mimics the studied ensembles of inter-electrode conductive bridges. Larger values of $K_{p}$ should lead to larger values of the first derivative $d\bar{U}/dt$ for the same values of $\bar{U}$. Note that there is a certain correlation between the shifts of the datasets towards large values of $d\bar{U}/dt$ and $N_{b}$, $\langle w_{b}\rangle$ values characterizing the samples under study. The parameter $\Sigma_{b}=K_{b}{\langle w\rangle }^{2}N_{b}$ can be considered as a measure of the total cross-section of inter-electrode bridges in the sample. Accordingly, the average current density in the examined samples is proportional to the ratio $I/\Sigma_{b}$. Figure 11 displays the dataset shifts ${\left(d\bar{U}/dt\right)}_{offset}$ along the $d\bar{U}/dt$ axis (Figure 10) against $\Sigma_{b}$; the shifts were estimated with respect to the reference point ($d\bar{U}/dt$ = 1.0×10^−3^ V/s; $\bar{V}$ = 10 V, see Figure 10b). The values of ${\left(d\bar{U}/dt\right)}_{offset}$ are associated with the introduced rate parameter $K_{p}$ up to a constant factor (see Equation (4)); thus, it can be concluded that lower average current densities in the samples at stage III cause larger rates of their approach to the percolation threshold. This conclusion may seem rather trivial, but the established features of the influence of the geometry of systems, described in terms of the parameter $\Sigma_{b}$, on the decay of ohmic conductivity can be useful for a deeper understanding of the behavior of such systems near the percolation threshold. In particular, one of these features is a close-to-linear relationship between $\Sigma_{b}$ and $K_{p}$ (Figure 11).

It is interesting to compare the obtained value of the critical exponent for ohmic conductivity with a variety of currently known similar values, obtained mainly using statistical modeling or other theoretical approaches. Numerous theoretical studies of percolation in 3D resistive grids carried out since the nineties of the last century gave the value of the critical conductivity exponent close to 2.0 (see, e.g., [26,27,28]). These values significantly exceed the value established in accordance with the method under discussion. On the other hand, modeling of percolation in two-dimensional lattices leads to the values of the critical conductivity exponent of the order of 1.3 (see, e.g., [29,30]). When comparing the value obtained from our experimental results and the critical exponents obtained for the model 3D and 2D percolation systems, it must be taken into account that the conducting structures (inter-electrode bridges) studied in our case are rather transitional between two-dimensional and three-dimensional conducting systems (the form factor $K_{b}$ is very small). It should also be noted that a rigorous theoretical analysis of the “conductor-insulator” transition using the transfer matrix approach was carried out in [31] and gave a value of the critical conductivity exponent of the order of 1.54 ± 0.08, which is fairly close to our result. 

### 4.3. Comments on DC Conductivity and Permittivity of the Studied Systems

We consider the following qualitative model of conductivity evolution in the ensembles of inter-electrode bridges under the influence of direct current:(1)during a short-term stage I (see inset in Figure 7a), the sets of conduction channels (percolation clusters) are formed in the inter-electrode bridges (Figure 12);(2)a long-term quasi-stationary stage II is characterized by a gradual decrease in ohmic conductivity of the bridges (see dataset 1 in Figure 9) due to a decrease in the number of conduction channels previously formed at stage I; this effect can be considered in terms of accumulating decrease in the concentration of electrons due to their capture by deep traps in anatase nanoparticles during electron transport in the bridges;(3)as the number of conduction channels in the bridges approaches a critical value corresponding to the percolation threshold, a rapid decrease in ohmic conductivity and, accordingly, an abrupt increase in the voltage drops across the studied samples occurs (stage III).

It can be assumed that the characteristic formation time $t_{f}$ for the sets of conduction channels in the bridges at stage I should correlate with the characteristic time of electron transfer through the bridges. This assumption makes it possible to roughly estimate the electron mobility $\mu_{e}$ in the studied structures of close-packed anatase nanoparticles. 

Accordingly, we can write the following approximate expression for the relationship between $t_{f}$ and the inter-electrode distance $d_{ie}$:(5)$$d_{ie}\approx \frac{\mu_{e}}{d_{ie}}\int_{0}^{t_{f}}U\left(t\right)dt.$$

As $t_{f}$, we take the time interval when the derivative $dU\left(t\right)/dt$ at stage I decreases (see inset in Figure 8a) to a level of 0.05 from the initial value. Thus, the electron mobility in the studied structures can be roughly estimated as:(6)$$\mu_{e}\approx \frac{d_{ie}^{2}}{\int_{0}^{t_{f}}U\left(t\right)dt}.$$

Analysis of initial fragments of $U\left(t\right)$ dependencies (stage I) for a group of five samples with various $N_{b}$ and $\langle w_{b}\rangle$ but the same inter-electrode distance $d_{ie}$ ≈ 44.9 μm gave a range of $\int_{0}^{t_{f}}U\left(t\right)dt$ as ≈(6.18 ± 3.69) V × s. Accordingly, $\mu_{e}$ for the studied samples were roughly estimated as ≈(3.72 ± 2.16) × 10^−6^ cm^2^/V × s. It should be noted that this estimation approach is simplified and does not take into account a variety of influencing factors (e.g., non-uniformity of the electric field in the inter-electrode space, and so on). Therefore, strictly speaking, its applicability is limited to estimates of the order of magnitude. Nevertheless, the obtained value of the electron mobility satisfactorily agrees with the value $\mu_{e}$ ≈ 5.0 × 10^−6^ cm^2^/V × s for the nanoporous anatase, obtained using the measurements of the photoresponse under laser irradiation and presented in [32]. The specimens studied in this work were obtained by annealing a mixture of 16-nm particles of anatase and turpentine oil on glass substrates and were characterized by the anatase volume fraction of about 50%. 

Based on the obtained approximate value of $\mu_{e}$ and typical voltage drops $U_{b}$ at the beginning of a quasi-stationary stage II (Figure 7a), we can roughly estimate the initial concentration $n_{e}$ of conduction electrons in the formed bridge-like structures. Depending on $N_{b}$ and $\langle w\rangle$, these voltage drops for the samples of the second group range from 1 to 3 V. The value of $n_{e}$ can be expressed as $n_{e}\approx Id_{ie}/\left(N_{b}K_{b}{\langle w\rangle }^{2}e\mu_{e}U_{b}\right)$, where e is the electron charge. Accordingly, $n_{e}\approx$ (1.31 ± 0.52) × 10^18^ cm^−3^. Taking into account the factor ψ of filling the volume of bridge-like structures with nanoparticles, we can estimate electron concentration in the nanophase. The approximate value ψ≈ 0.45 was obtained on the basis of volumetric estimations and measurements of $N_{b}$ and $\langle w\rangle$. As a result, we arrive at the following estimate for the anatase nanophase: $n_{e,\;nph}=n_{e}/\psi \approx$ (2.92 ± 1.15) × 10^18^ cm^−3^. This value can be compared to the published experimental data on the concentration of conduction electrons in the undoped anatase (see numerous references and corresponding Table 2, p. 2109 in [33]). It should be noted that, depending on the preparation technique, these data are strongly scattered in the range from ≈1.0 × 10^16^ cm^−3^ to ≈9 × 10^19^ cm^−3^, though, most results fall within the range of ≈3.2 × 10^17^ cm^−3^ to ≈2.0 × 10^19^ cm^−3^. Thus, the obtained value of $n_{e,\;nph}$ seems quite reasonable.

It is interesting to estimate an average number of conduction electrons per one nanoparticle; in our case, this value occurs approximately equal to ${\tilde{N}}_{ce}\approx$ 24. On the other hand, each nanoparticle is characterized by a certain number of deep traps capable of arresting conduction electrons. According to the photoconductivity data for the nanostructured anatase, the estimated number of traps $N_{tr}$ per one nanoparticle with an average diameter of 16 nm is of the order of 28 [34]; so a large value is reasonably explained by a great number of surface states in anatase nanoparticles. In our case, the average number of traps per particle presumably exceeds this value due to a larger average surface of particles. It is obvious that the ratio between $N_{tr}$ and ${\tilde{N}}_{ce}$ has a crucial influence on the degradation of ohmic conductivity in the process of electron transfer in bridge-like structures. A detailed quantitative analysis of this influence is beyond the scope of this work and is the object of further study.

Consideration of a small but systematic increase in the capacitance of samples resulting from formation of ensembles of inter-electrode bridges (see Section 2) makes it possible to estimate the real part of their initial effective permittivity $\varepsilon_{ef}$. A rigorous quantitative analysis of the influence of $\varepsilon_{ef}$ on $C_{sample}$ is a complicated problem, which is far beyond the scope of this work. Therefore, we applied an approximate approach based on the assumption of a close-to-linear dependence of the capacitance increment $C_{sample}-C_{substrate}$ on the ratio of the area covered by the bridges ($S_{b}$) to the total area covered by the electrodes ($S_{e}$). Within this approach, the relative capacitance increment $\eta =\left(C_{sample}-C_{substrate}\right)/C_{substrate}$ can be expressed as $\eta \approx \left(\varepsilon_{ef}-1\right)\left(S_{b}/S_{e}\right)\psi_{h}$. In this expression, the factor $\psi_{h}$ is equal to the ratio of the average height of the bridges to the height of electrodes and takes into account a partial filling of the inter-electrode space along the height with the nanoparticles. For a confidence level of 0.9, the ratios $\left(S_{b}/S_{e}\right)$ for the samples of the first group are in the range from 0.00524 to 0.01212, and the mean value is ≈0.0087. Similarly, the $\psi_{h}$ factor is characterized by the range from 0.0514 to 0.0801 and a mean value of 0.0658. With an average value of η approximately equal to 0.03, we obtained the effective permittivity of the examined anatase bridges as ≈56.0 ± 17.4. Note that such high permittivity values for the anatase-based structures are not surprising. To verify the obtained data, low-frequency permittivity (in the range from 1 Hz to 10^3^ Hz) of the layers of densely packed nanoparticles (the same product #637254 of Sigma Aldrich Inc. (St. Louis, MO, USA) as applied for the sample preparation) was measured using the Novocontrol Alpha AN impedance measuring system. The layers were tightly pressed up to the values of the anatase volume fraction of the order of 0.7. The obtained value of the real part of low-frequency permittivity is approximately equal to 70. Applying the Maxwell Garnet model of an effective medium [35] and considering the anatase phase in the prepared layers as the matrix substance with air inclusions, we can recover the permittivity of the bulk anatase as ≈115. Evaluation of effective permittivity for the substance of bridges with 0.45 volume fraction of anatase carried out using the Maxwell Garnet model gave $\varepsilon_{ef}\approx$ 41. This estimate is remarkably less than the above presented mean value (≈56) resulting from the capacitance increment analysis. However, it should be noted that systematic errors of the Maxwell Garnet model rise when the volume fractions of the matrix and inclusions are comparable and their permittivity values strongly differ from each other. In addition, taking into account an abundance of assumptions in the capacitance analysis, we can suggest a satisfactory agreement between these data.

It should be noted that the effect of an anomalous increase in the permittivity of percolation systems near the percolation threshold discussed in a number of works (see, e.g., [36,37,38]) was not observed in our experiments. On the contrary, at the end of the first runs, when the ohmic conductivity of the examined samples decreases over short time intervals by two orders of magnitude or more (Figure 9), there is a significant (up to 2–3 times) decrease in η. Accordingly, the effective permittivity of the inter-electrode bridges falls by the same number of times. This effect is a direct consequence of the localization of a significant number of conduction electrons on the traps, which leads to a significant decrease in the polarizability of nanoparticles and, accordingly, of the entire substance forming the bridges. A similar effect associated with the localization of photoinduced electrons on the traps during a long-term laser action on anatase nanoparticles in the fundamental absorption band was discussed in [22]. 

### 4.4. Spectral Properties of the Noise of Conduction Current

Equation (2) can be rewritten in the following form:(7)$$I=I_{cond}\left(t\right)+C_{tot}\frac{dU\left(t\right)}{dt}.$$

Here, $I_{cond}\left(t\right)$ is an instantaneous value of conduction current associated with a transfer of conduction electrons between the systems of electrodes of opposite signs. Differentiating Equation (7) under the condition of constant total current (dI/dt=0), we obtain:(8)$$\frac{dI_{cond}\left(t\right)}{dt}=-C_{tot}\frac{d^{2}U\left(t\right)}{dt^{2}}.$$

Applying the Fourier transform to both parts of Equation (8), we obtain the following expression:(9)$$\int_{-\infty }^{\infty }\frac{dI_{cond}\left(t\right)}{dt}\exp\left(-i\omega \;t\right)dt=-C_{tot}\int_{-\infty }^{\infty }\frac{d^{2}U\left(t\right)}{dt^{2}}\exp\left(-i\omega \;t\right)dt$$

Here we can use the following fundamental property of the Fourier transformation:(10)$$\int_{-\infty }^{\infty }f^{\left(n\right)}\left(t\right)\exp\left(-i\omega \;t\right)dt={\left(i\omega \right)}^{n}\int_{-\infty }^{\infty }f\left(t\right)\exp\left(-i\omega \;t\right)dt,$$
where $f^{\left(n\right)}\left(t\right)$ is the n-order derivative of a time-dependent function $f\left(t\right)$. Accordingly, Equation (9) is reduced to the following form:(11)$$i\omega \int_{-\infty }^{\infty }I_{cond}\left(t\right)\exp\left(-i\omega \;t\right)dt=C_{tot}\omega^{2}\int_{-\infty }^{\infty }U\left(t\right)\exp\left(-i\omega \;t\right)dt$$

Calculating the squares of the moduli of both parts of Equation (11) and taking into account that the squares of the moduli of the Fourier transforms of $I_{cond}\left(t\right)$ and $U\left(t\right)$ determine the corresponding spectral densities:(12)$$S_{I_{cond}}\left(\omega \right)={\left|\int_{-\infty }^{\infty }I_{cond}\left(t\right)\exp\left(-i\omega \;t\right)dt\right|}^{2};\;S_{U}\left(\omega \right)={\left|\int_{-\infty }^{\infty }U\left(t\right)\exp\left(-i\omega \;t\right)dt\right|}^{2},$$
we finally arrive at the following relation between the spectral densities of the conduction current and voltage drop fluctuations:(13)$$\omega^{2}S_{I_{cond}}\left(\omega \right)=C_{tot}^{2}\omega^{4}S_{U}\left(\omega \right).$$

Thus, the spectral density of conduction current fluctuations can be recovered from the spectral density of voltage drop fluctuations by applying the renormalization factor $C_{tot}^{2}\omega^{2}$. The power spectra of the voltage drop fluctuations $S_{U}\left(\omega \right)$ typically exhibit a decay, which is close to the power law $S_{U}\left(\omega \right)\propto \omega^{-\gamma^{\prime }}$ in the frequency range from 0.01 Hz to 1 Hz (see Figure 8b). Accordingly, the spectral exponent of conduction current fluctuations $\gamma =2-\gamma^{\prime }$ is in the range of (−1.0 ÷ −0.5) at the quasi-stationary stage and decreases down to (−1.5 ÷ −2.0) at the final stages of the first runs. The behavior $S_{I_{cond}}\left(\omega \right)\propto \omega^{-1}$ is associated with a classical flicker noise observed in a variety of conducting systems. At the same time, a remarkably high noise level for the examined systems should be noted; the normalized root-mean-square values of conduction current fluctuations at the quasi-stationary stage are typically in the range from ≈5 × 10^−3^ to ≈1.2 × 10^−2^. This feature is presumably due to the quasi-stationary dynamics of ensembles of conducting channels in all bridges. Such dynamics manifest themselves in the opening of new and blocking of some of the existing channels with minor changes in their total number.

A decrease in the spectral exponent γ of conduction current fluctuations down to the values close to −2 indicates the emergence of a new mechanism that controls a transfer of mobile charge carriers at the stage of significant conductivity decay. Moreover, considering a significant increase in the root-mean-square values of $U_{f}\left(t\right)$ at this stage (Figure 8a), it can be assumed an appearance of avalanche breakdowns in groups of mutually connected previously blocked channels with a rapid increase in the voltage drop across the samples. 

## 5. Conclusions

Thus, the observed features of electron transfer in bridge-like disordered ensembles of anatase nanoparticles demonstrate the achievement of a percolation threshold upon prolonged exposure to direct current and an extremely low rate of degraded conductivity recovery after the termination of the current action. These features can be interpreted as a manifestation of a “semiconductor-insulator” transition in these structures and are due to the depletion of ensembles of conduction electrons captured in the process of transfer through the structure by deep traps in anatase nanoparticles. This transition manifests itself in the limited ranges of the number of bridges and their average width, which determine a total cross-section of the conducting structure. The established value of the critical conductivity exponent for the studied structures has an intermediate value between theoretical values for three-dimensional and two-dimensional percolation systems. This statement seems reasonable since the form factor of inter-electrode bridges created during the experiment is quite small and, accordingly, the formed structures can be considered as transitional between two-dimensional and three-dimensional conducting systems.

For the examined samples, there are no manifestations of theoretically predicted and, in some cases, experimentally observed effects of a significant increase in the system permittivity in the region of the “conductor–insulator” transition. On the contrary, according to our estimates, there is a systematic decrease in the permittivity of the bridge material near the percolation threshold, due to the localization of the main fraction of mobile carriers in the traps and, accordingly, a decrease in the system polarizability in the constant electric field.

A remarkable feature of conductivity fluctuations in the studied systems as they approach the percolation threshold is a decrease in the spectral exponent, which characterizes a power-law decay in the spectral density of fluctuations, to the values close to −2. This feature indicates an increasing influence of avalanche-like local charge transfers in the groups of previously blocked conduction channels on the integral charge transfer when the threshold is approached.

The authors believe that the presented results and the discussed techniques will be useful in the further design and characterization of dispersed partially conductive nanomaterials for electronic and sensor applications.

## Figures and Tables

**Figure 1 nanomaterials-13-01490-f001:**
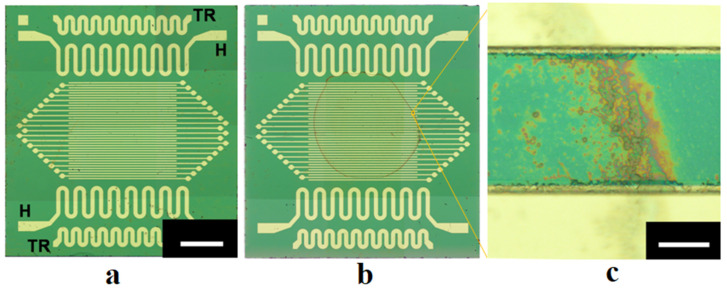
(**a**) The image of a substrate (the second group) with an interdigital electrode system used in the experiments; the white bar corresponds to 1.5 mm; (**b**) A substrate with “sown” anatase nanoparticles dominative forming a ring-like dispersive structure along the border of the applied volume of suspension; electrodes in the images (**a**,**b**) are not interconnected; (**c**) The image of an arbitrarily selected inter-electrode bridge; the white bar corresponds to 40 μm.

**Figure 2 nanomaterials-13-01490-f002:**
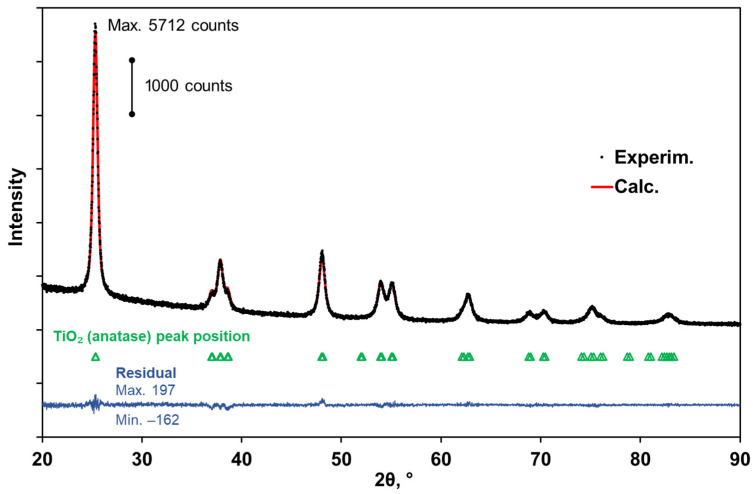
X-ray diffraction patterns for TiO_2_ (anatase) nanopowder: experimental (black dots), calculated (red), and difference curve (blue). Statistical correspondence between the model and experimental data (in terms of statistical estimates): weighted profile factor Rwp = 3.52, reduced χ2 = 0.88, the goodness of fit GoF = 0.94.

**Figure 3 nanomaterials-13-01490-f003:**
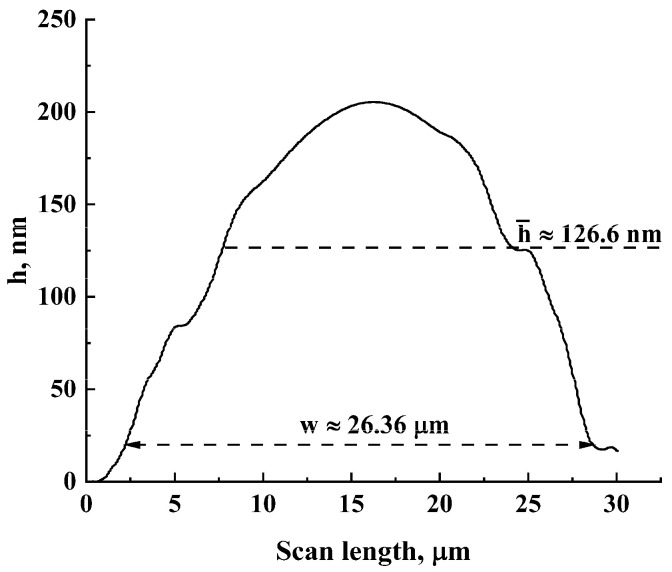
A typical smoothed profilogram for an arbitrarily chosen inter-electrode bridge scanned in the central part along electrode directions.

**Figure 4 nanomaterials-13-01490-f004:**
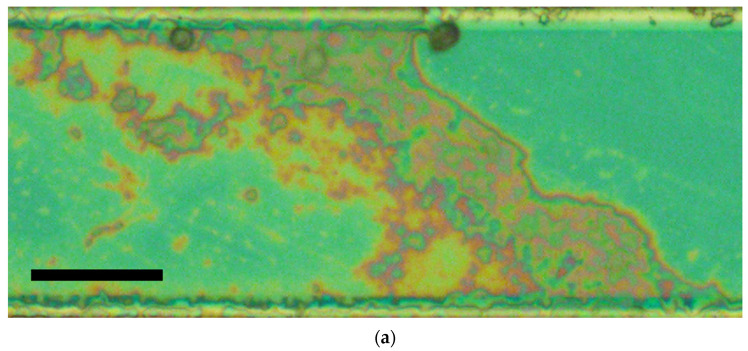
Estimation of the bridge width; (**a**) The initial image fragment; black bar corresponds to 20 μm; (**b**) The binarized image of the inter-electrode bridge.

**Figure 5 nanomaterials-13-01490-f005:**
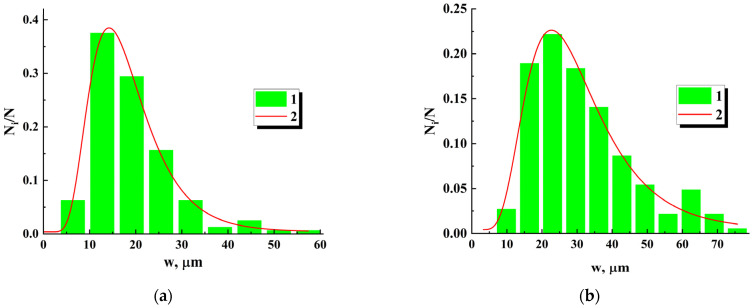
Examples of the histograms of the bridge widths (1) and corresponding lognormal approximations (2) for two examined samples. The approximation parameters are: (**a**) $\rho_{0}$ = 0.0039 ± 0.00633; $\sigma_{w}$ = 0.41832 ± 0.01684; $w_{c}$ = (16.935 ± 0.251) μm; $\langle w\rangle$ ≈ 18.5 μm; (**b**) $\rho_{0}$ = 0.00418 ± 0.00931; $\sigma_{w}$ = 0.45103 ± 0.03612; $w_{c}$ = (27.826 ± 0.964) μm; $\langle w\rangle$ ≈ 30.80 μm.

**Figure 6 nanomaterials-13-01490-f006:**
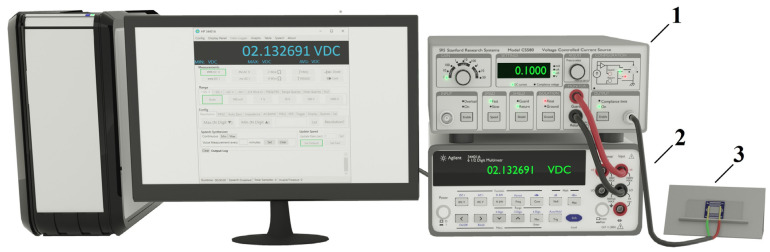
The experimental setup for studying conductivity evolution in the ensembles of anatase inter-electrode bridges; 1—voltage-controlled current source; 2—multimeter; 3—sample under study.

**Figure 7 nanomaterials-13-01490-f007:**
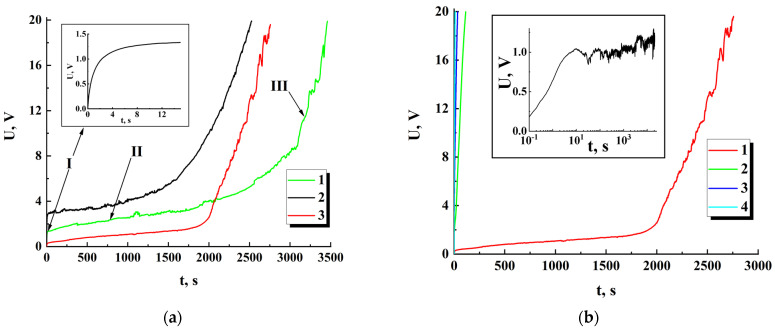
(**a**) Dependencies of the voltage prop on the time lapse along the first run for the samples with various $N_{b}$ and $\langle w\rangle$; 1—$N_{b}$ = 93, $\langle w\rangle$ ≈ 25.6 μm; 2—$N_{b}$ = 160, $\langle w\rangle$ ≈ 18.5 μm; 3—$N_{b}$ = 63, $\langle w\rangle$ ≈ 40.4 μm; the samples (1, 2) belong to the second group (see Section 2), and 3 belongs to the first group; the inset displays the $U\left(t\right)$ dependence for sample 1 at the beginning of the first run; (**b**) Evolution of the $U\left(t\right)$ dependencies for the sample with $N_{b}$ = 63 and $\langle w\rangle$ ≈ 40.4 μm in the sequence of runs (from 1 to 4); the inset displays the $U\left(t\right)$ dependence for the sample with a large amount of the inter-electrode bridges ($N_{b}$ = 376, $\langle w\rangle$ ≈ 29.2 μm).

**Figure 8 nanomaterials-13-01490-f008:**
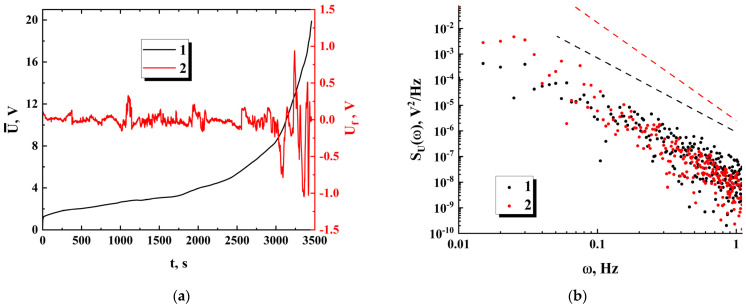
(**a**) The trend (1) and fluctuation (2) components of the $U\left(t\right)$ dependence at the first run for the sample with $N_{b}$ = 93 and $\langle w\rangle$ ≈ 25.6 μm (#1 in Figure 7a); (**b**) Fragments of the non-smoothed power spectra of the fluctuation component (#2 in Figure 8a) at the stages II and III; the dashed black and red lines indicate a power-law decrease in the spectral density and serve as guides for the eye; the sampled intervals in the time domain are (1000 ÷ 1300) s for (1) and (3100 ÷ 3400) s for (2); the values of $\gamma^{\prime }$ defined by the slopes of the trend lines are ≈ 2.9 (1) and ≈ 3.8 (2).

**Figure 9 nanomaterials-13-01490-f009:**
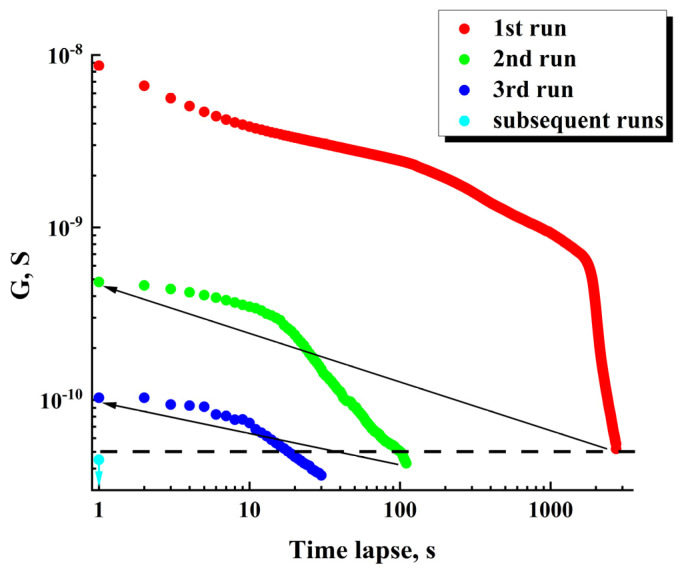
Degradation of ohmic conductivity in the sequence of runs; the initial data used for recovery are presented in Figure 7b.

**Figure 10 nanomaterials-13-01490-f010:**
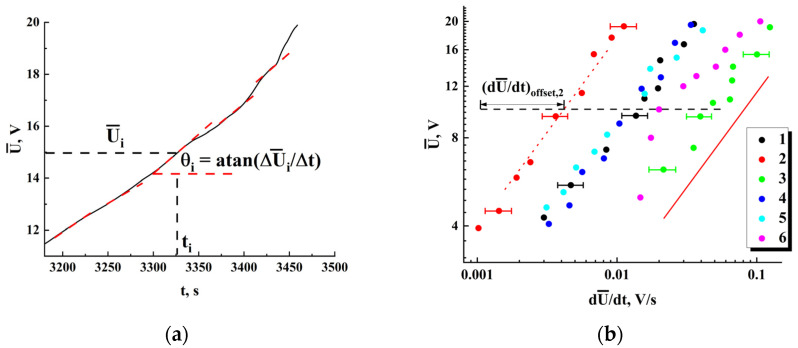
(**a**) Estimations of $\bar{U}$ and $d\bar{U}/dt$ from the trend line $\bar{U}\left(t\right)$; a set of dashed red segments illustrates random variations in the local values of the derivative $d\bar{U}/dt$, causing the data scatter in Figure 11b; (**b**) The values of $\bar{U}$ against $d\bar{U}/dt$ at stage III (see Figure 8a) for the samples with various $N_{b}$ and $\langle w\rangle$; 1 (the second group)—$N_{b}$ = 134, $\langle w\rangle$ ≈ 22.7 μm; 2 (the second group)—$N_{b}$ = 160, $\langle w\rangle$ ≈ 18.5 μm; 3 (the second group)—$N_{b}$ = 185, $\langle w\rangle$ ≈ 30.8 μm; 4 (the second group)—$N_{b}$ = 203, $\langle w\rangle$ ≈ 20.9 μm; 5 (the second group)—$N_{b}$ = 93, $\langle w\rangle$ ≈ 25.6 μm; 6 (the first group)—$N_{b}$ = 63, $\langle w\rangle$ ≈ 40.4 μm; the solid red line indicates a common power-law trend in the behavior of all datasets; selectively shown error bars indicate uncertainties in the estimates of $d\bar{U}/dt$ (see Figure 11a) and correspond to the confidence level of 0.9.

**Figure 11 nanomaterials-13-01490-f011:**
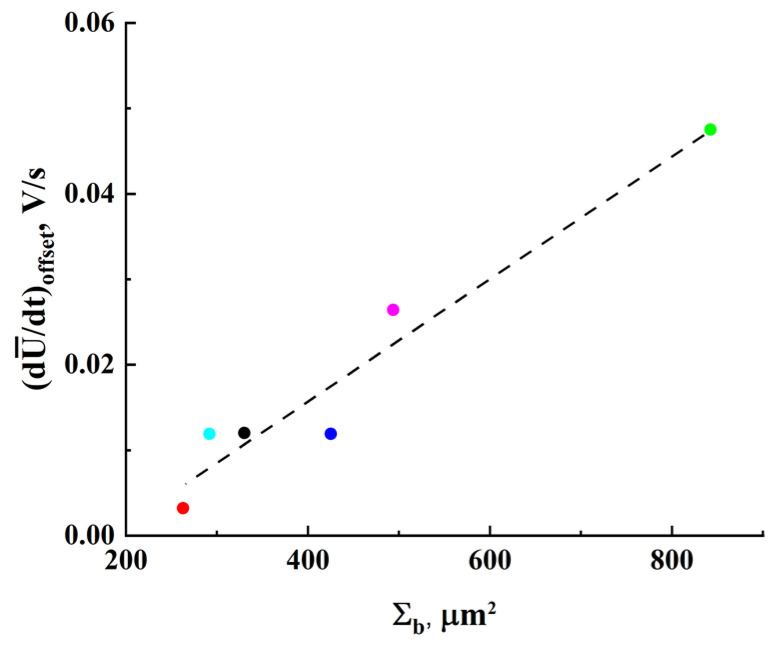
The ${\left(d\bar{U}/dt\right)}_{offset}$ values against the $\Sigma_{b}$ parameter. The coloring of the markers corresponds to those used in Figure 10b. The dotted line is a linear fit ${\left(d\bar{U}/dt\right)}_{offset}\approx -0.0129+7.2\times 10^{-5}\times \Sigma_{b}$ ) with $R^{2}\approx$ 0.94.

**Figure 12 nanomaterials-13-01490-f012:**
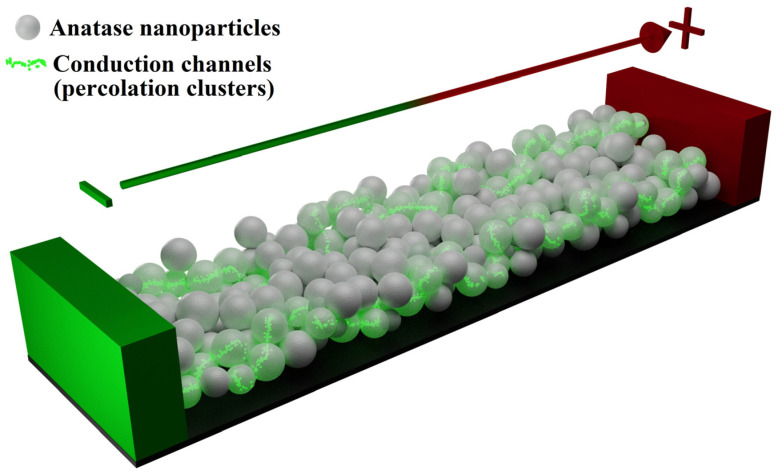
A model of electron transport through an inter-electrode bridge.

## Data Availability

Raw data will be provided by the corresponding author upon request.

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
