# Peer review of "Semiconductor-to-Insulator Transition in Inter-Electrode Bridge-like Ensembles of Anatase Nanoparticles under a Long-Term Action of the Direct Current"

_nanomaterials, 2023, doi:10.3390/nano13091490_

Round 1

Reviewer 1 Report

The authors presented the ohmic conductivity degradation in the ensembles of nanostructured anatase bridges under long-term effect of direct current. The results show that the ohmic conductivity degradation is irreversible, which are due to accumulated capture of conduction electrons by deep traps in anatase nanoparticles. A critical exponent for ohomic conductivity is 1.597 at the final stage, which agrees with the percolation systems. Here are some questions for the authors before this article can be completely accepted in the Journal. The problems and suggestions are listed as follows.

(1) Results and discussion can be combined in the revised paper.

(2) The units of the horizontal and vertical coordinates should be centered.

(3) P3, line 104, what is meaning of the product #637254? P3, line 136, what is meaning of “10-9÷10-8 S”?

(4) P7, line 199, “1.0·10-9A” should be revised “1.0 x10-9 A”

(5) The English of manuscript must be improved. We strongly suggest that you obtain assistance from a colleague who is well-versed in English or whose native language.

(6) Other important reports (J. Colloid Interface Sci., 2021, 601: 209-219; ACS Sustainable Chem. Eng., 2020, 8: 2707-2715) about semiconductor nanoparticles for other applications are also suggested to be included, which make the introduction more meaningful. 

The English of manuscript must be improved. We strongly suggest that you obtain assistance from a colleague who is well-versed in English or whose native language.

Author Response

The authors are grateful to the reviewer for valuable comments and suggestions that help improve the quality of the manuscript. Most of these comments and suggestions have been taken into account in the revised manuscript. Our responses, along with comments, are listed below.

  1. “Results and discussion can be combined in the revised paper”.

Of course, the "Results" and "Discussion" sections can be combined; however, such a combined section would be too long and overloaded with data and associated analysis. Moreover, each of these sections is additionally divided into several subsections. Therefore, we decided to keep the original structure of the manuscript, corresponding to the recommended article template for the "Nanomaterials".

  1. “The units of the horizontal and vertical coordinates should be centered.”

This was done for Fig. 2; Figure 3 with uncentered units was removed on the recommendation of another reviewer. On other figures, all units are centered.

  1. “P3, line 104, what is meaning of the product #637254? P3, line 136, what is meaning of “10-9÷10-8 S”?”.

The description of the used nanopowder is made more clear (see the green-marked additional text on page 3); the corresponding link is given in the list of references ([23]). The "10-9÷10-8 S" text fragment has also been corrected (see page 3).

  1. “P7, line 199, “1.0·10-9A” should be revised “1.0 x10-9 A”.

The recommended revisions were made throughout the manuscript text.

  1. “The English of manuscript must be improved. We strongly suggest that you obtain assistance from a colleague who is well-versed in English or whose native language”.

This comment seems somewhat debatable. Of course, the corresponding author responsible for the manuscript preparation is not a native English speaker and writer. However, he has a fairly successful and extensive experience in publishing scientific articles in English-language international journals. In addition, before submitting the manuscript, it was literary edited by our colleague, a professional English teacher with many years of experience in training and teaching in English-speaking countries. By the way, this is the first time we are faced with such a low assessment of our quality of English writing. However, after making revisions, the manuscript was additionally proofread and slightly edited by another English-experienced colleague.

  1. “Other important reports (J. Colloid Interface Sci., 2021, 601: 209-219; ACS Sustainable Chem. Eng., 2020, 8: 2707-2715) about semiconductor nanoparticles for other applications are also suggested to be included, which make the introduction more meaningful”.

That was done; see additional text fragment on page 1 and corresponding references [13] and [14] in the revised manuscript.

Reviewer 2 Report

In this article, Dmitry A. Zimnyakov et al., provided a comprehensive transport measurement and insightful discussion of the semiconductor-to-insulator transition in the solution-cast anatase nanoparticles. I agree with the authors that this work shall be useful for research committees related with electronic devices, particularly for those who focus on solution-processable nanomaterials. Therefore, I think this paper shall be publishable in Nanomaterials. Some minor concern or suggestion shall be considered for possible improvement:

1   1. One of my main concerns comes from the calculation for channel/bridge width and length in page 4, although the authors provides details description of microscopic image treatment as well as lognormal approximation in line 149-171.

(a)   Coffee ring effect does trend to form concentrated aggregation at the boundary of droplet, how can the authors exclude the possible connection or weak connection of anatase nanoparticles beyond the scope of microscopic image, particular considering the fairly long channel width in printed electrode (~several millimeters in Figure 1a), can the author clarify this point?

(b)   The anatase solution is not so uniformly dispersed on the substrate, a hydrophilic treatment like plasma, or UV-ozone can partially improve this issue, the authors shall consider it in subsequent work, after all, a uniform bridge is a very important issue to guarantee the repeatability and reliability of the transport measurement. Or the author shall consider post-deposition of electrode upon well-defined or characterized nanomaterials.   

(c)   Following last concern, how is the repeatability in this work, the semiconductor-to-insulator transition tendency between different device with various length, width or height, can the author comment on this point?

2   2.   Solution processable nanomaterials towards printed electronic devices represent an promising approach for the development of flexible, multifunctional and highly-integrated electronic systems, I also suggest some related papers for a possible discussion or perspective in this work. (Nature Materials 20.2 (2021): 181-187. Chemistry of Materials 30 (8), 2742-2749. Nature Reviews Materials 7.3 (2022): 217-234. )

3.   Some ways of expression are a bit weird to me, like page 3 line 136, “10-9  10-8 S ” means “10-9 to 10-8 S”? like page 3 line 144 “Kb  (4.8 ± 0.32)·10-3” shall generally be presented as “Kb  (4.8 ± 0.32) × 10-3

4.      Figure 3 does not reveal much useful information, I think this image shall be deletable.

the English and gramma is relatively very clear

Author Response

The authors are grateful to the reviewer for valuable comments and suggestions that help improve the quality of the manuscript. Most of these comments and suggestions have been taken into account in the revised manuscript. Our responses, along with comments, are listed below.

  1. “One of my main concerns comes from the calculation for channel/bridge width and length in page 4, although the authors provides details description of microscopic image treatment as well as lognormal approximation in line 149-171.

(a)   Coffee ring effect does trend to form concentrated aggregation at the boundary of droplet, how can the authors exclude the possible connection or weak connection of anatase nanoparticles beyond the scope of microscopic image, particular considering the fairly long channel width in printed electrode (~several millimeters in Figure 1a), can the author clarify this point?

(b)   The anatase solution is not so uniformly dispersed on the substrate, a hydrophilic treatment like plasma, or UV-ozone can partially improve this issue, the authors shall consider it in subsequent work, after all, a uniform bridge is a very important issue to guarantee the repeatability and reliability of the transport measurement. Or the author shall consider post-deposition of electrode upon well-defined or characterized nanomaterials.  

(c)   Following last concern, how is the repeatability in this work, the semiconductor-to-insulator transition tendency between different device with various length, width or height, can the author comment on this point?”

The reviewer is certainly right that the considered technique used for the formation of interelectrode bridge-like jumpers does not provide a uniform distribution of anatase nanoparticles in the interelectrode gaps (in particular, due to the coffee ring effect noted by the reviewer). On the other hand, this work is aimed not at the development of the technology for the synthesis of uniform coatings consisting of deposited nanoparticles. It is focused on the study of the fundamental features of irreversible percolation in low-dimensional dispersive structures of semiconductor nanoparticles. The obtained results show that despite the pronounced randomness of the formed bridge ensembles, which leads to a significant spread in their initial conductivities, their behavior when approaching the percolation threshold is characterized by some general features. These features are discussed in detail in the work.

It should also be noted that during the microscopic inspection, all interelectrode spaces were examined, and not only the areas associated with the “coffee ring”. A direct confirmation of the fact that bridges were taken into account over the entire surface covered by the electrodes, are typically the greater reported numbers of bridges compared to the maximal possible number of bridges in the “coffee ring” zone for a system of 19 pairs of interdigital electrodes (76). Indeed, the reported values are 63, 93, 134, 168, 185, and 203; only the sample with   = 63 is associated with the “coffee ring” conductance, whereas the other samples have remarkable contributions from local bridges inside the “coffee ring”. As an example, the fragment of the gray-scale image below (see the attached word file) shows an “internal” bridge, which was taken into account during the inspection.

Among the variety of bridges analyzed during the microscopic inspection, not a single one was found that completely covered the inter-electrode gap.  The feature with a length of several millimeters, which the reviewer drew attention to (“the fairly long channel width in printed electrode ~several millimeters in Figure 1a”, this is a Fig. 1b, sorry), is presented in the image fragment below (see the attached word file). It can be clearly seen that nanoparticles were deposited onto the electrode surface (white color) but not onto the inter-electrode gap (gray color).  Accordingly, we believe that a thorough microscopic inspection of all the studied samples provided reliable information about the number and ensemble-averaged width of conducting channels.

  1. “Solution processable nanomaterials towards printed electronic devices represent an promising approach for the development of flexible, multifunctional and highly-integrated electronic systems, I also suggest some related papers for a possible discussion or perspective in this work. (Nature Materials 20.2 (2021): 181-187. Chemistry of Materials 30 (8), 2742-2749. Nature Reviews Materials 7.3 (2022): 217-234. )”

That was done; see additional text fragment on page 1 and corresponding references [9,15,16] in the revised manuscript.

  1. “Some ways of expression are a bit weird to me, like page 3 line 136, “10-9 10-8 S ” means “10-9 to 10-8 S”? like page 3 line 144 “Kb ≈ (4.8 ± 0.32)·10-3” shall generally be presented as “Kb ≈ (4.8 ± 0.32) × 10-3””.

The recommended revisions were made throughout the manuscript text.

  1. “Figure 3 does not reveal much useful information, I think this image shall be deletable.”

This was done.

Reviewer 3 Report

0. General Aspects

-Please revise the manuscript looking for any typographic and grammatical error.

2. Materials and Methods

2.1.  -Authors are encouraged to describe the conditions used for the collection of the images (Figures 1 and 6) together with the imaging technique/s used in this work. Software or procedures used for the data processing and analysis of the images are also recommended to be mentioned.

2.2.  - Lines 248-250 and Figure 8. It would be recommended to include in the Figure 8 a record of the U(t) evolution for a representative sample with Nb > 300 supporting the comments included in the text. The trend highlighted by the authors does not seem to be fulfilled at Nb < 160, accordingly to the results shown in figure 8. In case any randomness in the results is present and common, this should be advised by the authors.

4. Discussion

4.1.Section 4.4. and equation 8. Authors have been very clear in the explanations and deduction of almost any previous equation or expression included in the work. Following such coherent argumentation line, it is suggested to further clarify the deduction of the equation 8 from eq. 7, either by including additional information or citing references to help guiding the reader for a more straightforward or comprehensive deduction.

The quality and clarity of the language is quite good. However, please revise the manuscript looking for any typographic and grammatical error.

Author Response

The authors are grateful to the reviewer for valuable comments and suggestions that help improve the quality of the manuscript. Most of these comments and suggestions have been taken into account in the revised manuscript. Our responses, along with comments, are listed below.

  1. “Please revise the manuscript looking for any typographic and grammatical error.”

This was done.

  1. “Authors are encouraged to describe the conditions used for the collection of the images (Figures 1 and 6) together with the imaging technique/s used in this work. Software or procedures used for the data processing and analysis of the images are also recommended to be mentioned.”

An additional paragraph describing the used microscopic technique and imaging conditions was included in the manuscript (see green-marked text fragment on pages 3-4). Regarding the software, we used a specially developed Matlab program to count the pixels in the black-and-white bridge images. A related comment is included in the text on page 4.

  1. “Lines 248-250 and Figure 8. It would be recommended to include in the Figure 8 a record of the U(t) evolution for a representative sample with Nb > 300 supporting the comments included in the text. The trend highlighted by the authors does not seem to be fulfilled at Nb < 160, accordingly to the results shown in figure 8. In case any randomness in the results is present and common, this should be advised by the authors”.

This was done. See the inset in Fig. 7, b and the corresponding figure caption in the revised manuscript.

  1. “Section 4.4. and equation 8. Authors have been very clear in the explanations and deduction of almost any previous equation or expression included in the work. Following such coherent argumentation line, it is suggested to further clarify the deduction of the equation 8 from eq. 7, either by including additional information or citing references to help guiding the reader for a more straightforward or comprehensive deduction.”

This was done. A derivation of an expression describing the relationship between the spectral densities of the conduction current and voltage drop is included in the manuscript (see formulas (7-13) and supporting text on page 16 of the revised manuscript. 

Reviewer 4 Report

This article is comprehensive, logically organized, and contains valuable information on the intercalation effects on the semiconductor-to-insulator transition in inter-electrode bridge-like ensembles of anatase nanoparticles under a long-term action of the direct current. The authors did excellent research on the experimental study and interpretation of this phenomenon in stochastic low-dimensional anatase nanostructures, which are random sets of local interelectrode bridges and are subjected to long-term exposure to direct electric current. The authors demonstrated that the observed features of electron transfer in the bridge-like disordered ensembles of anatase nanoparticles demonstrate the achievement of a percolation threshold upon prolonged exposure to direct current and an extremely low rate of degraded conductivity recovery after the termination of the current action. The submitted manuscript has significant scientific insights and the conclusions are soundly supported by the experimental data. Therefore, the present submission should be published in the future edition in the Special Issue of Properties and Potential Applications of Nanoparticles of the Nanomaterials.

Abstract: The results of experimental studies of ohmic conductivity degradation in the ensembles of nanostructured anatase bridges under the long-term effect of direct current are presented. Stochastic sets of interelectrode partially conducting bridges consisting of close-packed anatase nanoparticles were formed by the seeding particles from drying aqueous suspensions on the surfaces of silica substrates with interdigital platinum electrodes. Multiple-run experiments at room temperature have shown that ohmic conductivity degradation in these systems is irreversible. It is presumably due to the accumulated capture of conduction electrons by deep traps in anatase nanoparticles. The scaling analysis of voltage drops across the samples at the final stage of degradation gives a critical exponent for ohmic conductivity as 1.597. This value satisfactorily agrees with the reported model data for percolation systems. At an early stage of degradation, the spectral density of conduction current fluctuations within the frequency range of 0.01 Hz - 1 Hz decreases approximately as, near the percolation threshold, the decreasing trend changes. This transition is interpreted in terms of an increasing contribution of blockages and subsequent avalanche-like breakdowns of part of the local conduction channels in the bridges into electron transport near the percolation threshold.

Author Response

The authors are grateful to the reviewer for the high appreciation of their work.
